# Estimation of Cooling Rate of High-Strength Thick Plate Steel during Water Quenching Based on a Dilatometric Experiment

**DOI:** 10.3390/ma16134792

**Published:** 2023-07-03

**Authors:** Hyo-Haeng Jo, Kyeong-Won Kim, Hyungkwon Park, Joonoh Moon, Young-Woo Kim, Hyun-Bo Shim, Chang-Hoon Lee

**Affiliations:** 1Korea Institute of Materials Science, Changwon 51508, Republic of Korea; jhh4857@kims.re.kr (H.-H.J.);; 2Department of Materials Convergence and System Engineering, Changwon National University, Changwon 51140, Republic of Korea; 3Hyundai Steel Company, Dangjin 31719, Republic of Korea

**Keywords:** thick plate steel, water quenching, cooling rate, microstructure, hardness, dilatometry

## Abstract

The microstructure and hardness along the thickness direction of a water-quenched, high-strength thick plate with a thickness of 40 mm were investigated with three specimens from the thick plate: surface, 1/4t, and 1/2t (center) thickness, and the phase transformation behavior of the thick plate according to the cooling rate was analyzed through dilatometric experiments. Finally, the cooling rate for each thickness of the thick plate was estimated by comparing the microstructure and hardness of the thick plate along with the thickness with those of the dilatometric specimens. Martensite microstructure was observed on the surface of the water-quenched thick plate due to the fast cooling rate. On the other hand, an inhomogeneous microstructure was transformed inside the thick plate due to the relatively slow cooling rate and central segregation of Mn. A small fraction of bainite was shown at 1/4t thickness. A banded microstructure with martensite and bainite resulting from Mn segregation was developed at 1/2t; that is, the full martensite microstructure was transformed in the Mn-enriched area even at a slow cooling rate due to high hardenability, but a bainite microstructure was formed in the Mn-depleted area owing to relatively low hardenability. A portion of martensite with fine cementite at the surface and 1/4t was identified as auto-tempered martensite with a Bagaryatskii orientation relationship between the ferrite matrix and cementite. The microstructure and hardness as well as dilatation were investigated at various cooling rates through a dilatometric experiment, and a continuous cooling transformation (CCT) diagram was finally presented for the thick plate. Comparing the microstructure and hardness at the surface, 1/4t, and 1/2t of the thick plate with those of dilatometric specimens cooled at various cooling rates, it was estimated that the surface of the thick plate was cooled at more than 20 °C/s, whereas the 1/4t region was cooled at approximately 5~10 °C/s during water quenching. Despite the difficulty in estimation of the cooling rate of 1/2t due to the banded structure, the cooling rate of 1/2t was estimated between 3 and 5 °C/s based on the results of an Mn-depleted zone.

## 1. Introduction

As shipbuilding and construction have grown in scale, thick plate steels are increasingly in demand for their high strength and toughness [1,2,3,4,5,6,7,8,9,10,11,12]. In order for thick plate steels to have high strength and toughness, a thermo-mechanically controlled process (TMCP) is applied during hot rolling, or quenching and tempering (QT) heat treatment is introduced. For TMCP, strength and toughness can be improved by obtaining a low-temperature microstructure such as bainite by controlling the hot rolling temperature and applying accelerated cooling, which has a very high cooling capability, so there is no significant difference in microstructures in the thickness direction of a thick plate with a thickness of 40 mm. The other method for high strength and toughness in thick plates is QT heat treatment, which is aimed at obtaining a tempered martensite microstructure that provides an excellent combination of strength and toughness. At the surface area, a martensite microstructure can be easily obtained after water quenching due to the fast cooling rate, whereas the slow cooling rate in the center of the thick plate results in a transformation to bainite or ferrite. As the thickness increases, it is difficult to obtain a uniform microstructure in the thickness direction, leading to a complex microstructure composed of various phases such as martensite, bainite, and ferrite inside thick plates after water quenching. Although many researchers have tried to predict cooling rates in the thickness direction of thick plate steels during water quenching, a systematic analysis of cooling rates in thick plates has yet to be reported [13,14,15,16]. H. Wang et al. [13] studied the effect of cooling rate on microstructure and mechanical properties in low-carbon, low-alloyed steel plates using a dilatometric experiment. However, the authors focused on how the microstructure and mechanical property relationships of a low-carbon, low-alloyed steel are affected by phase transformations during continuous cooling, not on the estimation of the cooling rate in a thick steel plate. Lv Yanchun et al. [14] reported that temperatures in the core and surface of a steel plate with a thickness of 60 mm were measured by thermocouples embedded in the core and surface of the steel plate during normalizing, and the cooling rate was simulated through a finite element model using the measured temperature data. They did not consider the cooling rate of steel plate metallurgically by investigating microstructure, mechanical properties, etc. Therefore, in this study, our aim was to precisely identify the microstructure and hardness of high-strength thick plate steel with a thickness of 40 mm in the thickness direction as well as specimens from a high-strength thick plate controlled at various cooling rates by a dilatometric experiment. Finally, we estimated the cooling rates in the thickness direction of the thick plate during water quenching based on a comparison of the microstructure and hardness in both specimens from the thick plate and from the dilatometric experiments.

## 2. Experimental Procedure

Microstructures in the thickness direction of a water-quenched thick plate with a thickness of 40 mm and dilatometric specimens cooled with various cooling rates were carefully characterized by scanning and transmission electron microscopies, and the Vickers hardness of the thick plate and the dilatometric specimens were measured to evaluate mechanical properties.

### 2.1. Material Preparation and Characterization

The microstructure and hardness were investigated for high-strength thick plate steel with a thickness of 40 mm after austenitization at 900~930 °C for 60~90 min and water quenching (As-Quenched, AQ). The chemical composition is shown in Table 1. Note that details of the compositions of micro-alloying elements cannot be disclosed due to information security. The microstructure of the surface, 1/4t, and 1/2t in the thickness direction of the thick plate was observed using scanning electron microscopy (JSM-7001F, JEOL, Tokyo, Japan) and transmission electron microscopy (JEM-2100F, JEOL, Japan). The samples for scanning electron microscopy (SEM) were prepared using a 1% picral solution (1 g picric acid + 5 mL HCL + 100 mL ethyl alcohol) after mechanical polishing and disc samples with 3 mm diameter for transmission electron microscopy (TEM) were prepared by mechanical polishing to 70~100 μm thickness, followed by electro-chemical polishing using a twin-jet polisher with a solution of 10% perchloric acid and 90% methanol at −30 °C. Vickers hardness (FM-700, Future Tech, Kawasaki, Japan) was measured under a load of 1 kg for 10 s.

### 2.2. Dilatometric Experiment

Dilatometric specimens with dimensions of 3 mm diameter × 10 mm length were machined from the thick plate using a dilatometer (Dilatronic-III, Theta, San Jose, CA, USA). The heat treatment conditions for the dilatometric experiment are shown in Figure 1. After austenitizing at 950 °C for 5 min, cooling was performed at various cooling rates between 0.5 to 50 °C/s. It is difficult to find any evidence of undissolved cementites or heterogeneity of austenite during this austenitizing condition. A CCT diagram was prepared based on the dilatation results, the microstructure observation, and the Vickers hardness for the specimens controlled with various cooling rates. The specimens were observed by SEM (JSM-7001F, JEOL) and TEM (JEM-2100F, JEOL) and their Vickers hardness was measured under the same conditions as described above. It was confirmed that prior austenite grain sizes of the thick plate and the dilatometric specimen are very similar: 14.9 μm in the thick plate and 15.6 μm in the dilatometric specimen.

## 3. Results and Discussion

### 3.1. Investigation of AQ Specimens

Figure 2 shows SEM micrographs of the surface, 1/4t, and 1/2t in the AQ thick plate. The surface area shows a martensite microstructure. Martensite and a small fraction of bainite at austenite grain boundaries and a banded structure with martensite and bainite were observed at 1/4t and 1/2t, respectively. Bainite transformation occurs toward the center in thickness due to a slow cooling rate. It is well known that this banded structure at 1/2t is formed due to the segregation of alloying elements, especially Mn during slab solidification, resulting in Mn-enriched and Mn-depleted regions [17,18,19,20,21]. The Mn-enriched zone at 1/2t of the AQ specimen was transformed to martensite due to high hardenability even at a slow cooling rate, whereas a bainite transformation occurred in the Mn-depleted zone with relatively low hardenability. Table 2 presents the Vickers hardness according to the thickness. The hardness of the surface with martensite was approximately 500 Hv, and that of 1/4t, which contains martensite and a small amount of bainite, was 390 Hv. The hardness of the martensite and bainite in the banded structure of the 1/2t were measured as 507 Hv and 351 Hv, respectively. As a result, the Vickers hardness results are consistent with the microstructural change according to thickness. Figure 3 presents a high-magnification SEM image, a TEM bright field image, and the selected area diffraction pattern at 1/4t. In Figure 3a, martensitic lath and fine particles inside the lath were observed. Through a TEM analysis, this microstructure was identified as auto-tempered martensite with fine cementite, where a Bagayatskii orientation relationship between the ferrite matrix and fine cementite was found in Figure 3b,c. Auto-tempering of martensite is a phenomenon that can occur immediately after martensite transformation for steels with a relatively high martensite transformation start (Ms) temperature [22,23,24]. The Ms temperature of the AQ specimen is around 400 °C, which is relatively high, resulting in auto-tempering.

### 3.2. Dilatation Results of AQ Specimens

Figure 4 shows the dilatation behavior of an AQ specimen during heating and cooling. With a decreasing cooling rate, volume expansion occurs at a higher temperature during cooling, which is consistent with the previous studies [11,12,13]. Therefore, martensite can be transformed at a higher cooling rate, and bainite or ferrite can be transformed as the cooling rate becomes slower in the AQ specimen. The microstructure according to the cooling rate is presented in Figure 5. Full martensite was clearly shown at a cooling rate of 50 °C/s. At 20 °C/s, it was found that a very small amount of bainite was transformed. With a decreasing cooling rate to 3 °C/s, the amount of the transformed bainite increased and the fraction of martensite decreased. At cooling rates of 1 °C/s and 0.5 °C/s, the ferrite phase was found as well as bainite, but martensite was hardly observed. It is determined that auto-tempering of the dilatometric specimen cooled at various cooling rates occurred, as also observed in the AQ thick plate. Figure 6 shows the SEM and TEM micrographs of specimens cooled at 50 °C/s as an example of auto-tempering. It was confirmed again that AQ steel was auto-tempered due to its high MS temperature. Figure 7 shows the Vickers hardness of dilatometric specimens according to the cooling rate. As the cooling rate became slower, the Vickers hardness accordingly became lower, which is consistent with the microstructural change in Figure 5. Full martensite has a hardness of approximately 500 Hv. With an increasing fraction of bainite, the hardness decreases to 322 Hv for the specimen cooled at 3 °C/s. At much slower cooling rates of 1 °C/s and 0.5 °C/s, the hardness is below 300 Hv because of the transformation to ferrite, which is a soft phase.

A CCT diagram of the AQ specimen is presented in Figure 8, based on Figure 4, Figure 5, and Figure 7. It was found that full martensite was transformed at a cooling rate of 50 °C/s, bainite and martensite were transformed at a cooling rate in the range between 3~20 °C/s, and ferrite and bainite and no more martensite were transformed at a cooling rate lower than 1 °C/s. Bainite transformation start (Bs) and martensite transformation start (Ms) temperatures can be calculated by Equations (1) and (2), respectively [25,26,27,28]. The calculated Bs and Ms temperatures were 606 °C and 378 °C, respectively, which are similar to those in this CCT diagram.
[Bs, °C] = 830 − 270[C] − 90[Mn] − 70[Cr] − 37[Ni] − 83[Mo] (wt.%)(1)
[Ms, °C] = 561 − 474[C] − 33[Mn] − 17[Cr] − 17[Ni] − 21[Mo] (wt.%)(2)

Finally, the actual cooling rate for each thickness of the AQ thick plate during water quenching was estimated by comparing the microstructure and hardness between the AQ thick plate in the thickness direction and dilatometric specimens cooled at various cooling rates, as shown in Figure 9. The hardness of 500 Hv and martensite at the surface of the AQ thick plate show that the surface should be cooled at a cooling rate of 20 °C/s or more. The hardness of 390 Hv and microstructure of martensite and bainite at 1/4t were similar to those of the dilatometric specimens cooled at a cooling rate between 5 and 10 °C/s. In the case of 1/2t, it was difficult to estimate the cooling rate due to the banded structure, but if the results of an Mn-depleted zone are considered, the cooling rate of 1/2t was estimated between 3 and 5 °C/s.

## 4. Conclusions

In this study, the microstructure and hardness of an AQ thick plate with 40 mm thickness were investigated in the thickness direction. Dilatometric experiments with specimens from the AQ thick plate were conducted to verify the effect of the cooling rate on the microstructure and hardness and to make a CCT diagram of the AQ steel. Finally, we estimated the cooling rate of the AQ thick plate in the thickness direction during water quenching based on an investigation of the thick plate and dilatometric specimens. The surface of the thick plate showed a full martensite microstructure due to the fast cooling rate, and the bainite structure was partially transformed at 1/4t due to the slower cooling rate. At 1/2t, a banded structure with martensite and bainite was formed due to a hardenability difference resulting from the central segregation of Mn in the slab. By comparing the microstructure and hardness of dilatometric specimens cooled at various cooling rates, the cooling rates at the surface and at 1/4t of the thick plate during water quenching were more than 20 °C/s and between 5 and 10 °C/s, respectively. The cooling rate at 1/2t was estimated between 3 and 5 °C/s based on the results of an Mn-depleted zone. A portion of martensite in both the AQ thick plate and the dilatometric specimen was identified as auto-tempered martensite in which there is a Bagayatskii orientation relationship between the ferrite matrix and cementite.

## Figures and Tables

**Figure 1 materials-16-04792-f001:**
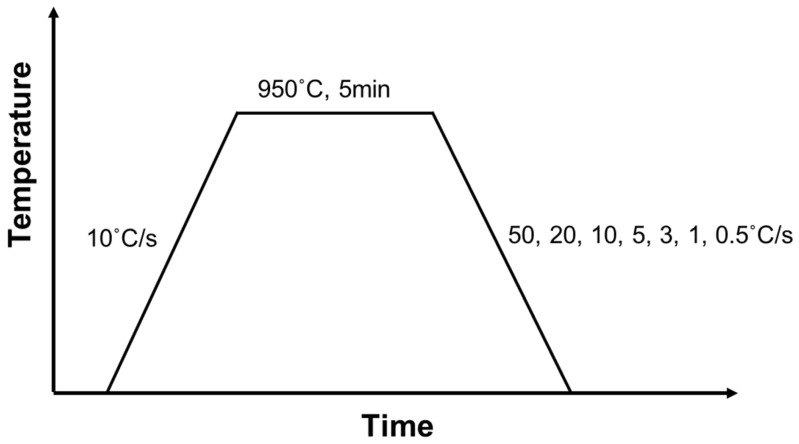
Heat treatment conditions for dilatometric experiment.

**Figure 2 materials-16-04792-f002:**
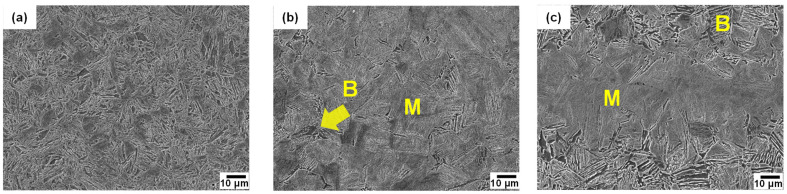
SEM micrographs of AQ specimen: (**a**) surface, (**b**) 1/4t and (**c**) 1/2t in thickness. (M: martensite, B: bainite).

**Figure 3 materials-16-04792-f003:**
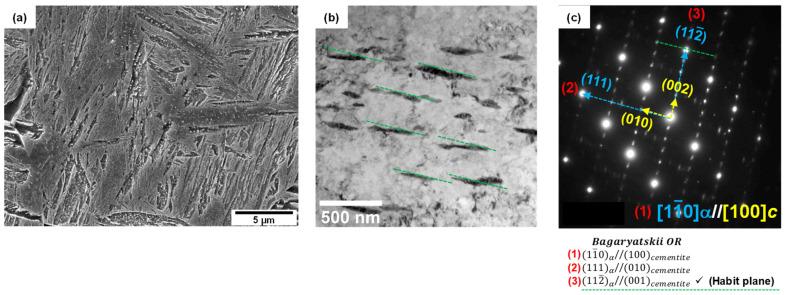
Auto-tempered martensite at 1/4t of AQ specimen: (**a**) SEM micrograph with high magnification, (**b**) TEM micrograph, and (**c**) selected area diffraction pattern showing Bagaryatskii orientation relationship between ferrite and cementite in auto-tempered martensite. (green line: habit plane).

**Figure 4 materials-16-04792-f004:**
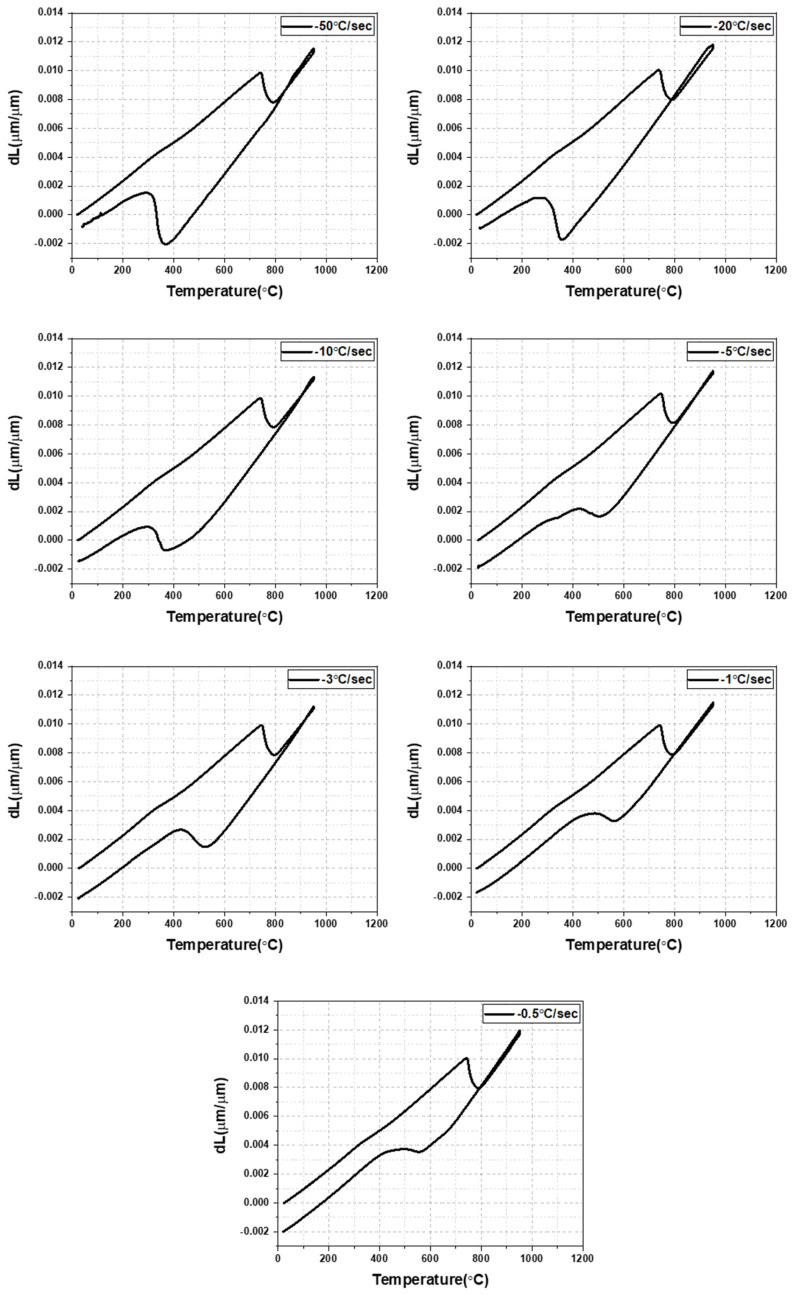
Dilatation results of AQ specimen according to cooling rate.

**Figure 5 materials-16-04792-f005:**
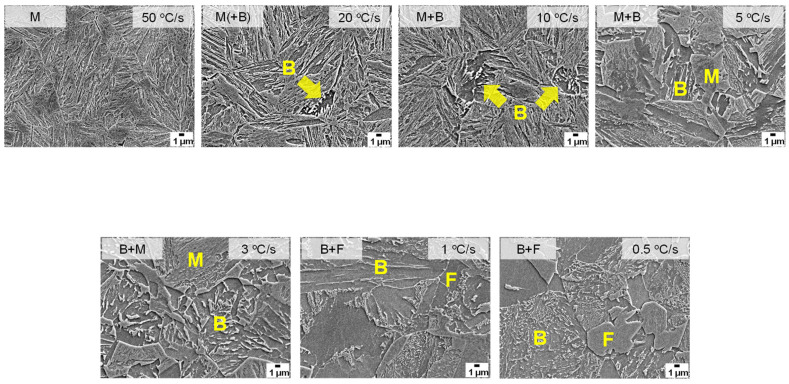
SEM micrographs of dilatometric specimens according to cooling rate. (M: martensite, B: bainite, F: ferrite).

**Figure 6 materials-16-04792-f006:**
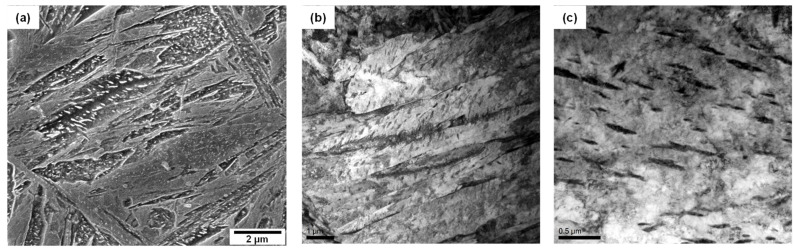
Auto-tempered martensite in dilatometric specimen cooled at 50 °C/s: (**a**) SEM micrograph with high magnification, (**b**,**c**) TEM bright field images.

**Figure 7 materials-16-04792-f007:**
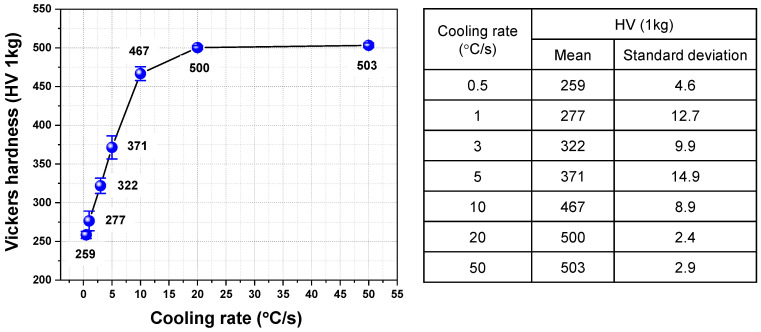
Vickers hardness of dilatometric specimens according to cooling rate.

**Figure 8 materials-16-04792-f008:**
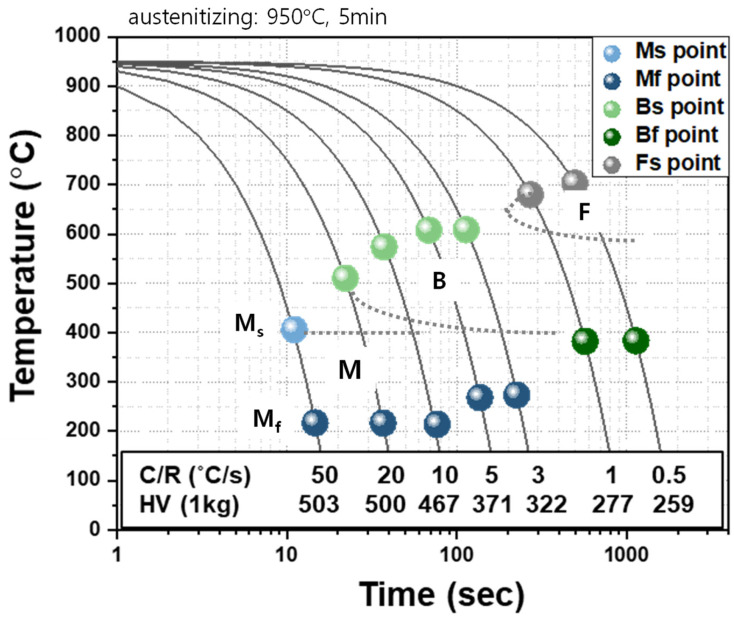
CCT diagram of AQ specimen based on a dilatometric experiment.

**Figure 9 materials-16-04792-f009:**
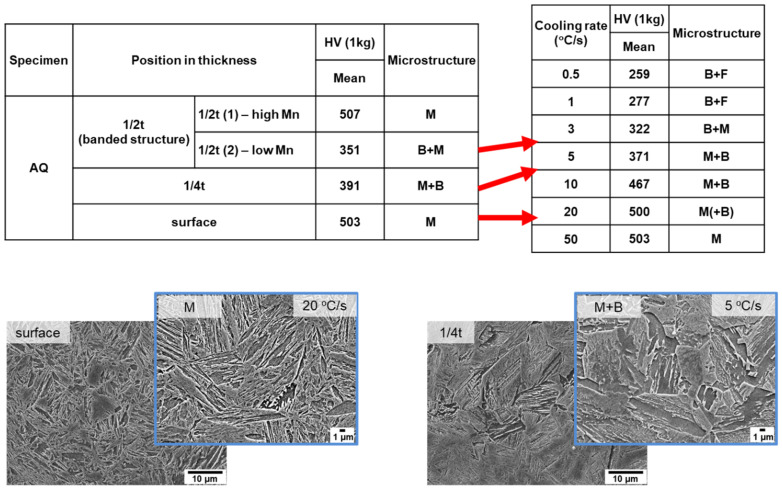
Estimation of cooling rate of AQ thick plate during water quenching by comparison of hardness and microstructure between AQ thick plate and dilatometric specimens.

**Table 1 materials-16-04792-t001:** Chemical composition of thick plate steel.

Specimen	Thickness (mm)	Chemical Compositions (wt%)				
		C	Si	Mn	V + Nb + Ti	Cu + Ni + Cr + Mo
AQ	40	0.28	0.3	0.7	0.05	1.6

**Table 2 materials-16-04792-t002:** Vickers hardness of AQ specimen.

Specimen	Position in Thickness		HV (1 kg)
			Mean (Standard Deviation)
AQ	1/2t (banded structure)	1/2t (1)—Mn-enriched zone	507 (±18.3)
		1/2t (2)—Mn-depleted zone	351 (±16.7)
	1/4t		391 (±17.1)
	surface		503 (±3.5)

## Data Availability

The data presented in this study are available on request from the corresponding author.

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
