# Peer review of "Estimation of Cooling Rate of High-Strength Thick Plate Steel during Water Quenching Based on a Dilatometric Experiment"

_materials, 2023, doi:10.3390/ma16134792_

Round 1
Reviewer 1 Report
this paper estimates the cooling rate of high-strength thick plate steel during water quanching based on a dilatometric experiment. The research is interesting and the paper is well written. There is one suggestion as below:
The figure 4 has too small text font, which should be large enough to read easily.
Author Response
Please refer to Response to comments.

Reviewer 2 Report
The paper is interesting and with good practical application of the results. It is not clear how the authors determine the cooling rate in the thick plate. The experimental work is clearly explained and logical and consistent. I have only few minor comments that must be addressed before the publication.
1. Please, give the explanation of the experimental approach in the beginning of the experimental part , not at the e not the introduction. It will improve the readability.
2. It is good also if a data for the variation of the cooling rate in this or similar steel grade can be shown for comparison.
3. Table one is not clear . Please, show information only for the elements that are really present in the steel. Do not leave empty spaces for the elements that are not present. If it is impossible to disclose the exact alloying element content provide them as sum. For example “sum of V+TI+Nb=XXXwt%)
4. Line 61: “Finally, we predicted the cooling rates in the thickness direction of the thick plate during water quenching based on a comparison of the microstructure and hardness in both specimens from the thick plate and from the dilatometric experiments.”
5. This is not exactly true. The authors do not predict the cooling rates. They estimate what the cooling rates were at different thicknesses of the plate based on the microstructures. Please correct.
6. Change Fig. 1. It is not serious to have a graph without a coordinate system, irrespectively that is self-evident what it represents. This is a paper in a scientific journal.
The language quality is at acceptable and easy -to read level.
Author Response
Please refer to Response to comments.

Reviewer 3 Report
The article is devoted to the investigations of microstructure and hardness changes along the thickness direction of high strength thick plate after a water quenching and additional dilatometry experiments with different cooling rates. The connection between the cooling rate and the formed steel microstructure at different depths from the surface is determined. The article is interesting. The research was done carefully, but the reviewer had some questions:
1. It is not clear from the introduction why this steel was chosen for research. What are its advantages compared to other steels of this class? Why is the research useful? Where can they find practical application?
2. Line 66: "The microstructure and hardness were investigated for high-strength thick plate steel with a thickness of 40 mm after water-quenching ...". What is the hardening temperature, holding time in the austenitic region? What are the standard (quenching and tempering) treatment parameters for the steel under study (temperature, time)?
3. What are the geometric dimensions of the investigated 40 mm plate (length, width)?
4. The authors indicate that there are areas enriched and depleted in manganese on ¼ of the plate thickness (Table 2). Has the elemental analysis been carried out in these areas? The steel contains only 0.7 wt.% manganese. What exactly is the difference in manganese content?
5. Figure 4 does not show the designation of the axes.
6. In a dilatometric study, exposure at 950 °C was only 5 minutes. This is probably not enough to homogenize austenite. What condition was the steel in before dilatometric testing?
7. The paper states that during dilatometric studies at a cooling rate of 50 °C/sec, self-tempering of martensite occurs. Was cementite not detected at other cooling rates?
8. When studying the microhardness of a 40 mm thick plate at ½ thickness, the authors separated the areas of bainite and martensite. In the study of samples after dilatometric tests with low cooling rates, such separation is not shown. Is the average value taken here (regardless of where the Vickers pyramid fell - in the bainite region or the martensite region)?
9. Figure 9 repeats other figures and tables.
Author Response
Please refer to Response to comments.

Reviewer 4 Report
Dear authors,
The paper is well written with significant novelty. On top of that, the paper highlights important and also new findings for the widely investigated research on the effects of cooling rates on the steel specimen. However, the authors need to check the short word used for example TEM and it is highly advisable to write the full name prior to use the short-term word. Overall, the paper can be accepted and published in its current form.
The quality of English language is highly acceptable, and the authors have done an excellent work on writing the paper using English language although the authors are not native speaker for the English language.
Author Response
Please refer to Response to comments.
